# MLPs for NLP: Towards Discovering Inductive Bias From Scratch

## Abstract

The recent rise of large language models has been fueled by scale. More data, more compute, and bigger models have consistently led to better performance. This scaling paradigm has been applied most notably to the transformer architecture, which is especially conducive to training parallelization and sequence modeling. In this work, we ask what happens if we apply the power of scale to the simplest possible class of neural network: the multi-layer perceptron (MLP). Specifically, we train MLPs to perform next-token prediction on billions of tokens of text. Indeed, their performance consistently improves with scale, though vanilla MLPs are still clearly inferior to transformers for this task, especially because their parameter count grows with the length of the input sequences. We then perform a mechanistic analysis of the trained models, and identify a consistent emergent structure: most neurons in the first hidden layer either perform arbitrary linear functions over a small look-back window, or low-frequency functions over the entire context. These neuron types recall $n$-gram and bag-of-words techniques from classical statistical language modeling. Using the discrete cosine transform, we define a unified way of reparameterizing these neuron types such that the number of parameters per neuron does not depend on the sequence length.

## 1 Introduction

It has been almost a quarter-century since neural language models were introduced in the seminal work of Bengio et al. (2000). The authors trained multi-layer perceptrons (MLPs) to perform next-word prediction on large text corpora, and found that the resulting models outperformed leading $n$-gram based models, which had been state-of-the-art since Shannon first introduced language modeling a half-century earlier (Shannon, 1951). MLPs, consisting of fully-connected layers of neurons, are the prototypical basic deep learning architecture (Rumelhart et al., 1986). In their largest experiment, Bengio et al. trained a width-120 MLP with a context length of 8 on a training set of 32 million words.

In this work, we train simple MLPs on language modeling, just like Bengio et al. (2000), but at a much larger scale—ReLU MLPs with up to 900 million parameters and 128 tokens of context, supervised on 10 billion tokens. As observed in other contexts (Hestness et al., 2017; Kaplan et al., 2020; Hoffmann et al., 2022; Bachmann et al., 2023), increases in scale lead to consistent improvements in performance. See Figure 1 for empirical scaling results and comparisons to the transformer architecture (Vaswani et al., 2017).

Recently, Bachmann et al. (2023) performed an extensive study of scaling MLPs in the vision domain, and some of our motivations parallel theirs. Much theoretical research in deep learning studies MLPs because of their simplicity, and we are interested in how much relevance this literature has for the large-scale language modeling setting. We also see our MLP experiments as a test of the limits of scaling: can an extremely generic architecture, given enough parameters and data, perform arbitrarily well at language modeling? Of course, there are reasons vanilla MLPs are not actually a practical architecture for language modeling. Most importantly, a given MLP can only support a fixed context length, and scaling the context length requires increasing the number of parameters.

But training MLPs also provides an opportunity. Because the architecture is so generic, we can think of it (informally) as a "blank slate" on which the learning process builds circuits for language modeling unconstrained by human-imposed inductive biases. If we are lucky, we might identify

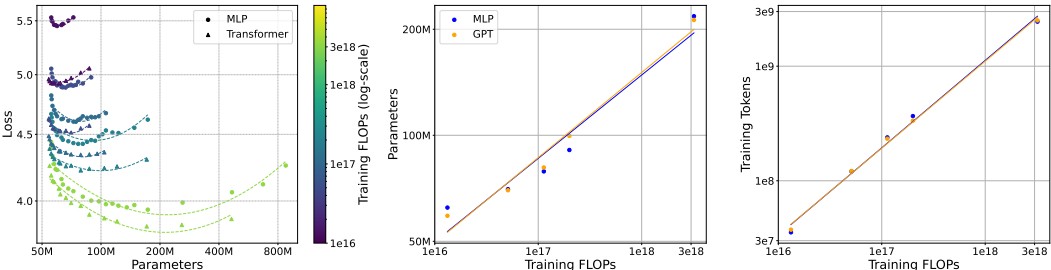

Figure 1: Following Method 2 from Hoffmann et al. (2022), we fix five compute budgets and vary the amount of parameters and the amount of training tokens within each compute budget. We then fit a parabola to each compute budget curve (left). We then take the minimum of each parabola and plot training FLOP vs optimal parameter count (center), training FLOP vs optimal token count (center).

emergent regularities that could be hard-coded into new architectures, potentially improving their language modeling inductive bias, computational efficiency, or interpretability.

We thus perform a mechanistic analysis of the neurons in the first hidden layer of our models. The results are striking: most neurons can be assigned to one of two types (see Figure 4). "$n$-gram" neurons perform an arbitrary linear function over the most recent $n - 1$ tokens for some small value of $n$, ignoring all of the preceding context. "Bag-of-words" neurons, meanwhile, apply a linear map to the average of all the input token embeddings; in fact, this linear map typically picks out a semantically coherent direction in token embedding space. This emergent structure echoes non-neural language modeling techniques which combine $n$-gram and bag-of-words statistics (Wallach, 2006). Although the proportion of neurons that fit neatly into these two categories increases during the course of training, at the end of training there remain other "hybrid" neurons which can be expressed as a linear combination of $n$-gram and bag-of-words neurons.

Directly inspired by this empirical finding, we introduce a new type of neuron, the *DCT (discrete cosine transform) neuron*, which is rich enough to express $n$-gram neurons and bag-of-words neurons as special cases, and has a number of parameters that does not depend on the sequence length. Specifically, we leverage the fact that both types of neurons have weight matrices that are restricted to some look-back window, and (for each embedding coordinate) are low-frequency functions (in comparison to the total context length) over that window.

In summary: We begin by studying the scaling behavior of vanilla MLPs in a simple language modeling setting in Section 2. We then perform an analysis of the neurons of the resulting model in Section 4, in which we introduce the $n$-gram and bag-of-words neuron types. We conclude by proposing a method of reparameterizing the MLP language model using the DCT, such that the number of parameters does not depend on sequence length (Section 5). We conclude with an overview of related work (Section 6 and a high-level discussion 7).

## 2 VANILLA MLP FOR SHORT-SEQUENCE LAST-TOKEN PREDICTION

While transformers have dominated the landscape of language modeling, we take a step back and ask: *With enough compute, can an MLP achieve comparable performance in language modeling to architectures with more inductive bias?* To study this, we simplify the language modeling task to a setting in which it is feasible to train and scale MLP models, as a case study in how neural networks model language when removing the impact of architectural biases.

We train a vanilla MLP architecture on short sequence language modeling. Specifically, our MLP architecture consists of an input embedding layer, three hidden layers, and output unembedding. We use LayerNorm (Ba et al., 2016) after linear layers for training stability, ReLU activations, and residual connections (He et al., 2015). To further simplify the language modeling task, we have the model predict only the last token of each input sequence. For our main experiments, we train on sequences of length 16 where the network sees the first 15 tokens and is tasked with predicting the last token; we count tokens by the number of *supervised output* tokens. An illustration of the MLP architecture is shown in Figure 2 where $T$ represents the sequence length, $d$ represents the embedding

dimension, and $w$ represents the width of the MLP. We choose this design of the MLP Block in order to allow residual connections between layers.

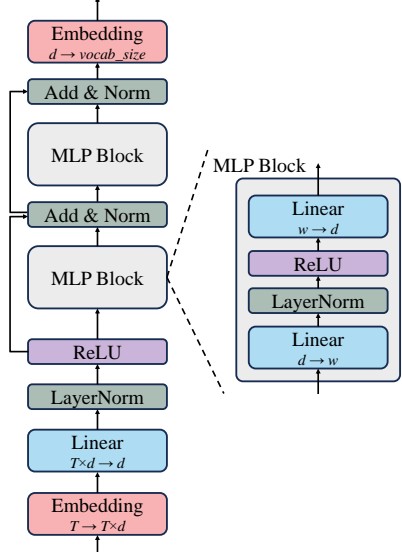

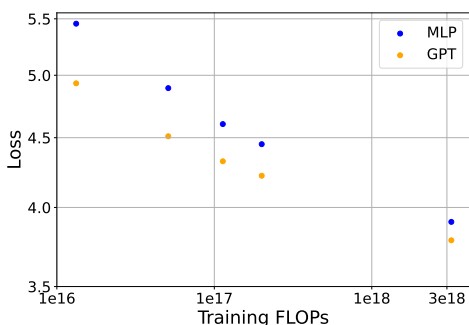

Figure 3: Training FLOPs vs optimal loss corresponding with Figure 1.

Figure 2: Illustration of the MLP architecture.

We compare our MLP architecture with a non-causal version of the GPT-2 architecture (Radford et al., 2019) where only the last token is supervised. Our GPT model is depth 2 with 16 attention heads. Both models have an embedding dimension of 512 and use AdamW (Loshchilov & Hutter, 2019) as the optimizer. Full training details are listed in Appendix B.

We train on the C4 (Raffel et al., 2023) split of the Dolma dataset (Soldaini et al., 2024) and we use disjoint sequences as training samples. The C4 dataset contains roughly 200 billion tokens of web content. We evaluate over a validation set containing a held-out set of data from the C4 corpus.

## 3 SCALING LAWS

We study how MLP models scale with increasing compute as compared to the transformer architecture on our language modeling task (Figure 1). Specifically, we follow Method 2 from Hoffmann et al. (2022) to examine the trade-off between increasing parameters and increasing training tokens within a fixed compute budget. We train models over a range of 50M to 900M parameters (including all embedding parameters) for five compute budgets between 1e16 and 3e18. We measure compute budget in terms of training FLOPs using the approximation $C = 6ND$ (Kaplan et al., 2020) for a compute budget $C$, number of parameters $N$ and number of supervised training tokens $D$. We set the cosine schedule to match the number of training steps for each run. We plot number of parameters (on a log scale) versus loss and fit a quadratic curve through the data points of each compute budget. We find that increasing depth does not make a large difference as compared to scaling width for the MLP models, so for simplicity we fix the models to have three hidden layers and scale only the width. We also fix depth for the transformer models, noting the finding from Kaplan et al. (2020) that scaling trends are reasonably consistent for depths of at least 2. We note that the MLP curves display a "bump"—a local maximum in the compute budget curves; this plausibly could be resolved with optimal hyperparameter tuning, but due to resource constraints we leave this to future investigation.

The minima of the fitted parabolas correspond to estimates of the compute-optimal parameter count $N_{opt}$ for each compute budget. We use these estimates to plot training FLOPs vs compute-optimal parameter count and training FLOPs vs compute-optimal token count $D_{opt}$ (Figure 1), as well as training FLOPs vs compute-optimal loss (Figure 3). While the transformers achieve consistently lower loss at the same compute level, the scaling trends are strikingly similar between architectures.

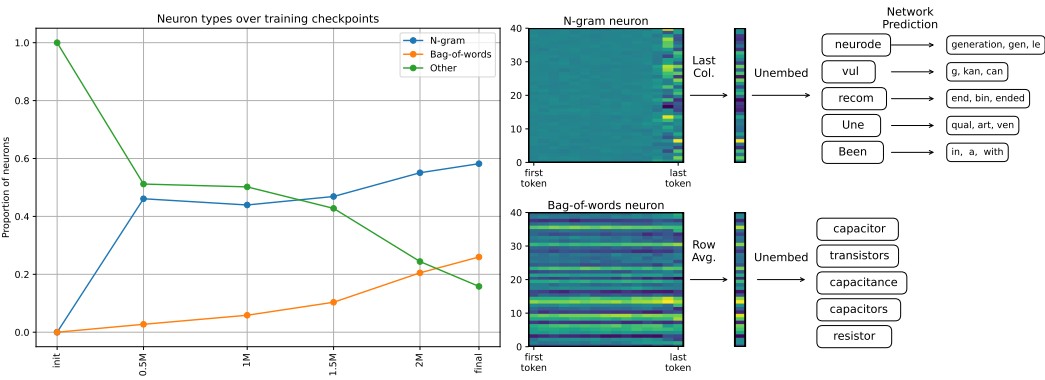

Figure 4: We train a width-16,384 vanilla MLP to predict the last token of 10 billion length-16 sequences of text. **Left:** Over training, increasingly many neurons can be categorized as $n$-gram neurons or bag-of-words neurons (see Section 4.1). **Right:** Weight matrices of an example $n$-gram neuron and a bag-of-words neuron, with corresponding keywords (see Section 4.2. The x axis is the time dimension and the y axis is a size-40 slice (for compactness) of the embedding dimension.

We fit power laws to obtain exponents $a$ and $b$ for $N_{opt} \propto C^a$ and $D_{opt} \propto C^b$. Even though the two architectures are very different, their exponents are essentially the same: for the MLP, $a = 0.24$ and $b = 0.76$, and for the Transformer, $a = 0.24$ and $b = 0.76$. We note that the true curves do not seem to be perfectly captured by a power law fit. We also note the difference from the exponents found in Hoffmann et al. (2022) ($a = 0.49$ and $b = 0.51$); the differences in behavior of the transformer model from our setting and Hoffmann et al. (2022) likely arises from our short-sequence last-token prediction setting, rather than the more standard longer-sequence next-token prediction setting.

These results suggest that despite being less computationally and data efficient, the MLP can become a successful language model when trained with sufficient compute, and indeed its qualitative scaling behavior is very similar to that of the GPT reference architecture.

## 4 ANALYSIS OF THE UNDERLYING ARCHITECTURE

We have shown that an MLP can be a non-trivial language model; now, we investigate what this model has learned. Since the MLP architecture has minimal built-in inductive bias, we are interested to see what structures emerge in the trained language model, as such structures can potentially inspire some "natural" inductive bias. In other words, *can what the MLP model learns inform how we build a more scalable architecture?* In particular, we wish to find a parameterization or sub-architecture of the MLP model with a number of parameters that does not depend on sequence length.

In this section, we analyze what structures emerge at the neuron level, and introduce a unifying view of these structures. In Section 4, we discuss a potential implementation of this unifying view such that the resulting model *does not depend on sequence length* in its number of parameters.

### 4.1 NEURON-TYPES ANALYSIS

To better understand the MLP language model, we analyze the individual neurons of the trained MLP model. In particular, we analyze the neurons of the first linear layer of the MLP mapping $(T \cdot d) \mapsto d$, where $T$ is the input sequence length and $d$ is the embedding dimension. We reshape the $(T \cdot d) \times d$ weight matrix to shape $(d, T, d)$. This allows us to look at neuron weights across both the time (sequence) and embedding dimensions. We focus on this first layer due to the fact that it retains the time dimension, which is lost in future layers of the network. We observe that each neuron either computes an arbitrary linear function over a small, consecutive span of tokens or a "smooth" function over all of the tokens. This is a surprising finding: even though we do not limit the expressive power of the neurons explicitly, they learn to compute functions that are restricted in their complexity. We call these two main groups of neurons *n-gram* neurons and *bag-of-words* neurons respectively. We choose these names due to the connection to the two corresponding classical NLP methods. It is

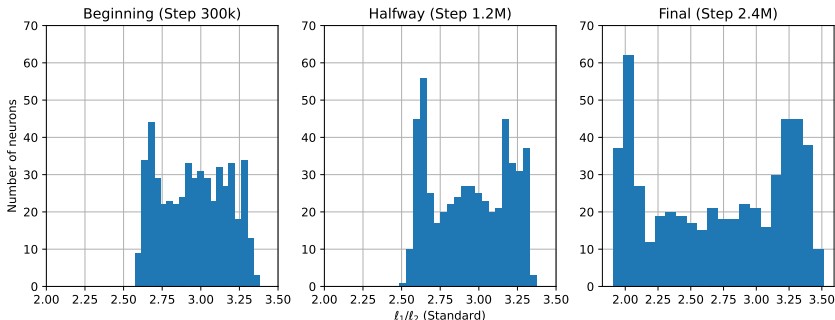

Figure 5: $\frac{\ell_1}{\ell_2}$ measured of a 16384-width MLP trained on 10b tokens at initialization (right), halfway through training (center), and at the end of training (left).

interesting to note that the MLP "discovered" these two unique methods for language processing: this recalls prior works such as Wallach (2006) which explicitly combine the power of $n$-gram and bag-of-words models. We qualitatively define each of these types as follows:

- $n$-**gram neurons**: Neurons that contain non-zero weights only in a small, consecutive span across the time dimension. Within this span, an arbitrary linear function is computed over the embedding dimension. Most of the neurons in this category focus on the most recent tokens.

- **Bag-of-words neurons**: Neurons that share weights over the time dimension, i.e. apply the same function to each token regardless of position.

We show examples of each type of neuron in Figure 4 and additional examples in Appendix F. In addition, we observe "hybrid" neurons that combine these two components, having an arbitrary function of the last few tokens and a shared weight for the earlier tokens.

Following these observations, we classify neurons to $n$-gram or bag-of-words using simple heuristics. For the $n$-gram neurons, we check that a high proportion of rows of the time dimension are close to zero and for the existence of a row of the time dimension with high norm, which indicates that the neuron has large weights only on a subsequence of the input context. For the bag-of-words neurons, we check for low variance across the time dimension. The exact formulations and thresholds are specified in Appendix H.

Using these heuristics, we plot the proportion of $n$-gram, bag-of-words, and other neurons over the course of training for a 16,384 width MLP trained on ten billion tokens (Figure 4). We observe that $n$-gram neurons emerge first, early in training, while the bag-of-words neurons evolve more gradually. We also study how the proportion of neuron types changes with model width, and we find that the proportions appear to converge for large enough models (Appendix E).

We further corroborate the existence of $n$-gram and bag-of-words neurons by measuring the ratio of the $\ell_1$ and $\ell_2$ norms over the time dimension. For each neuron $\mathbf{w} \in \mathbb{R}^{T \times d}$, we compute

$$\frac{\ell_1}{\ell_2} := \frac{1}{d} \sum_{i=1}^{d} \frac{\|w_{:,i}\|_1}{\|w_{:,i}\|_2}$$

In an idealized setting, note that a unigram neuron with non-zero weights only across a single index $t$ of time has $\frac{\ell_1}{\ell_2} = 1$, and a bag-of-words neuron with equal weight for each index of time has $\frac{\ell_1}{\ell_2} = \sqrt{T}$. Figure 5 shows this quantity for neurons of the 16384-width MLP trained on 10b tokens over three checkpoints. For this model, we have $T = 15$ and therefore $\sqrt{T} \approx 3.87$. In line with the idealized setting, we observe peaks at the extrema of the range of $\frac{\ell_1}{\ell_2}$ values, suggestive of $n$-gram neurons and bag-of-words neurons. We also observe how the categories as suggested by the $\frac{\ell_1}{\ell_2}$ measure disambiguate over the course of training; at initialization, the values are highly concentrated, while at the end of training we see a bimodal distribution emerge.

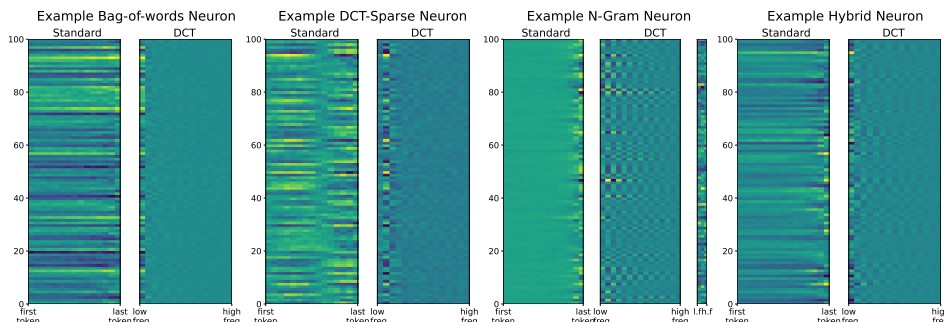

Figure 6: Examples weight matrices of individual neurons, paired with these matrices after application of DCT across the time dimension. The x axis is the time dimension and the y axis is the embedding dimension. The bag-of-words and DCT-sparse neurons use only the first few frequencies to represent the weights over the entire context. For the $n$-gram neuron, the first middle column shows the DCT over the entire context length, and the rightmost column shows the DCT over only the last three columns of the neuron weights.

## 4.2 A note on interpretability

Given that the neurons in this first layer operate directly on the embeddings of the input sequence, we are interested in exploring whether we can interpret what these neurons are doing at a semantic level, as has been done by prior work such as Elhage et al. (2021) for simple transformer language models. For the bag-of-words neurons, nearly the same vector of weights is applied to each unit of the time dimension. Given that this vector is of size $\mathbb{R}^d$, we study whether this vector has an interpretable meaning once unembedded. We take the average of the weights across time to obtain a single vector of dimension $d$. We then look at the top-5 closest words to this vector in token embedding space to understand the concepts associated with each bag-of-words neuron. We observe that the keywords associated with a given bag-of-words vector are often related concepts.

For the $n$-gram neurons, we take the weights of the most active row as measured by the $\ell_2$-norm and project it back into token space. We then run inference on our model with an input of all zeros besides the last token (which is the projected column). We then observe how the model completes this 2-gram. When unembedding the last column of the $n$-gram to get the 5-closest words as we do in the bag-of-words case, we find that these keywords themselves tend to be less correlated from a human-centered interpretability lens. However, these neurons often appear to be $n$-grams with $n > 1$, so it is possible that a single unit of the $n$-gram cannot be interpreted in isolation. An example of each neuron type is given in Figure 4, and additional examples are listed in Appendix G. We encourage future work in the interpretability of both the $n$-gram and bag-of-words neuron categories.

## 4.3 DCT analysis of neurons

Besides the dominant structures of $n$-gram and bag-of-words neurons, we additionally look at the remaining neurons to see if more minor structures emerge. In doing so, we observe a third type of neuron that resembles low frequency cosine waves across the time dimension. Thus, we apply the Discrete Cosine Transform as an empirical analysis tool in understanding structures that emerge in the frequency domain.

The Discrete Cosine Transform (DCT), closely related to the Discrete Fourier Transform but producing only real outputs, expresses a sequence as a sum of cosine waves of different frequencies. For a sequence $x_1, \ldots, x_{T-1}$, the DCT Type II is defined as $\text{DCT}_i^{(T)} = 2 \sum_{j=0}^{T-1} x_j \cos\left(\frac{\pi i(2j+1)}{2T}\right)$, up to a scaling factor depending on the normalization mode. The output $\text{DCT}_i^{(T)}$ denotes the $i^{th}$ DCT coefficient of a length-$T$ sequence. The first $T$ functions of the DCT form a basis for arbitrary sequences of length $T$.

We filter for neurons that are sparse in the DCT basis by checking that a high proportion of rows of the time dimension are close to zero and for the existence of a row of the time dimension with high

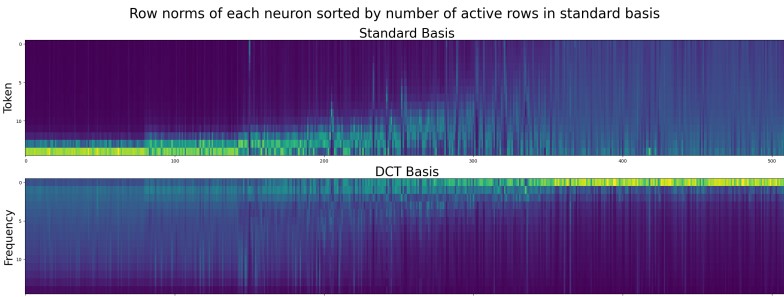

Figure 7: Row norms of each neuron sorted by number of rows of large weights in the standard basis. Each column depicts the row norm of a neuron. The top shows the row norms of the neurons in the standard basis and the bottom shows the row norms of the neurons in the same order after DCT is applied.

norm on the DCT of the neuron. We refer to this neuron type as the *DCT-sparse* neuron (Figure 6). Figure 6 shows the DCT for an example of each type of neuron. The bag-of-words neurons can be represented using essentially only the lowest frequency, which serves as an average over the time dimension. The DCT-Sparse neuron, which resembles a low frequency cosine wave, can similarly be represented using only the first few frequencies. The $n$-gram neuron, while not sparse if computing the frequency over the entire sequence length, can be restricted to a short sequence length to obtain a low-dimensional DCT representation. The "hybrid" neurons can be easily represented as a sum of its $n$-gram and bag-of-words components. Note that the bag-of-words neuron is a special case of the DCT-Sparse neuron that uses only the lowest frequency. When filtering for DCT-sparse neurons, we exclude previously categorized neuron types to keep these categories distinct for visualization purposes.

We empirically verify the emergence of DCT-sparse neurons. Figure 7 shows the row norm of each neuron, sorted by the number of rows with large weights in the standard basis (we define large weights as having a norm exceeding 0.01). This allows us to visualize the $n$-gram and bag-of-words neurons alongside their weights in the DCT basis. As shown in Figure 7, the bag-of-words neurons (at the right end of the plot) in general use only the lowest few frequencies in the DCT basis. The $n$-gram neurons, while using the higher frequencies, only use a small "look-back" window in the standard representation. Therefore we can restrict the $n$-gram window to a small look-back window to achieve a DCT representation with a small number of frequencies. The middle of the figure captures the "hybrid" neurons that interpolate between bag-of-words and $n$-gram neurons, with large weight values over a small window and a constant value over a wider range in the standard space, and a large value of the low frequency with some additional higher frequency information in the DCT space. We also consider the evolution of row norms in the standard vs DCT basis over time to track the emergence of sparse-DCT neurons (See Appendix J).

## 5    DCT-PARAMETERIZED LONG-CONTEXT MLP

In Section 2 we showed that MLPs can be trained for next-token prediction on short context lengths, but are hard to train with long context windows. However, when we study the weights of the neurons learned by the network, we observe that these neurons converge to particular low-complexity weights: high-frequency local functions ($n$-gram neurons) or low-frequency global functions (bag-of-words neurons). In this section we leverage these observations to construct a reparameterization of the MLP that satisfies two properties: 1) the number of parameters does not grow with the sequence length and 2) this module can still express the neuron types discovered in the previous section. To achieve this, we will replace the linear layer of the MLP by a *linear network*, with some learned parameters and some fixed linear computations.

We begin by describing DCT-Parameterized Neurons, a unification of both the $n$-gram and the bag-of-words neurons. Then, we will show how these can be used to construct a transformer-like causal language model, where we replace the attention blocks with the DCT-parameterized MLPs.

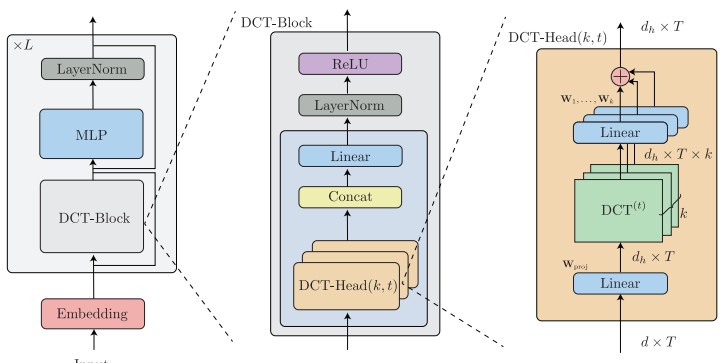

Figure 8: Illustration of DCT-MLP architecture

## 5.1 DCT-Parameterized Neurons

Recall that, as observed in Section 4, the neurons in the trained MLP learn to either ignore most of the context length, computing a function over the last few tokens, or integrate information from all tokens but compute a low-frequency function. In other words, each neurons can "choose" a look-back window-size $t$, that indicates how many tokens into the past can it depend on, but it is forced to use only $k$ frequencies. With this in mind, we define the DCT-Parameterized Neuron, which has two hyper-parameters $k, t$, satisfying $1 \leq k \leq t \leq T$, and learned parameters $\boldsymbol{w}_1, \ldots, \boldsymbol{w}_k \in \mathbb{R}^d$:

$$\text{DCT-Neuron}_{k,t}(\boldsymbol{x}_1, \ldots, \boldsymbol{x}_T; \boldsymbol{w}_1, \ldots, \boldsymbol{w}_k) = \sum_{i=1}^{k} \boldsymbol{w}_i^{\top} \text{DCT}_i^{(t)}(\boldsymbol{x}_{T-t}, \ldots, \boldsymbol{x}_T) \tag{1}$$

where $\text{DCT}_i^{(t)}$ indicates the $i^{th}$ frequency of the DCT operator applied over a sequence of length $t$ (where the operator is applied independently to each coordinate in $\mathbb{R}^d$). Observe that the DCT Neuron is simply a re-parameterization of the "standard" neuron and the number of parameters does not depend on the sequence length $T$; more notes on the DCT-Neuron are provided in Appendix C.

## 5.2 Causal Language Modeling with DCT-Parameterization

Our goal now is to define a causal language modeling architecture using DCT-neurons. Specifically, we will replace the causal attention heads of the transformer architecture by a causal DCT-head. Each head has two fixed hyper-parameters: the number of frequencies $k$ and the look-back window size $t$.

We then, similarly to the transformer, alternate between DCT layers and tokenwise MLP layers. The full architecture is described and illustrated in Appendix C..

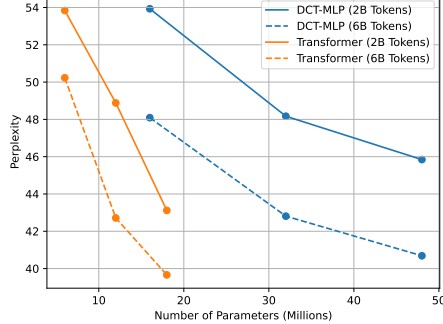

Figure 9: Comparison between the perplexity of DCT-MLP and a small transformer on C4.

**Results** We train the DCT-Parameterized MLP (DCT-MLP) on the C4 dataset (Raffel et al., 2023) on sequences of length 2,048. (Full experimental details are provided in Appendix C). Evidently, the Transformer is still more efficient in various aspects: it requires less depth, less data and less parameters to achieve the same performance. However, we note that, at least at this scale, by scaling the number of parameters and depth by around 3x, our MLP is able to match the performance of a small transformer. By letting the MLP train on 3x more data, this gap shrinks, where an MLP requires only 1.5x more parameters to achieve competitive performance. All in all, while the MLP is admittedly inferior to the transformer, we view these results as a positive evidence that a

reparameterized MLP can be scaled to perform language modeling on a long context lengths. We leave this as a preliminary study and encourage future work in designing new architectures leveraging these findings.

# 6  RELATED WORK

**Scaling laws**    Scaling laws for transformers are well-studied and demonstrate that transformers exhibit a power law in which performance increases with increasing compute (Kaplan et al. (2020), Hoffmann et al. (2022)). Additionally, Tay et al. (2022) studies the impact of inductive bias on scaling behavior and finds scaling laws vary from model to model. Bachmann et al. (2023) introduces scaling laws for MLPs on vision tasks. We extend this study to a simple language modeling setting.

**MLP-inspired architectures**    Several past works study the question of how to best encode token-mixing as an inductive bias, and in particular how self-attention can be replaced with simpler modules. Tolstikhin et al. (2021) introduce MLP-Mixer and Touvron et al. (2023) introduce ResMLP, both methods of applying alternating *channel-mixing* and *token-mixing* MLPs, primarily focusing on the image domain. Melas-Kyriazi (2021) conducts a similar study. In addition, Liu et al. (2021) introduces the gMLP architecture, which uses gating as an alternative to self-attention. All four works indicate that self-attention is sufficient but not necessary when given enough compute. Our work addresses a similar question, but we offer the novel approach of searching for this inductive bias by attempting to encode what the vanilla MLP learns.

**Fourier transform-based networks**    Several works study how to incorporate spectral information as an architectural bias. Rippel et al. (2015) and Chi et al. (2020) discuss Fourier-based methods for convolutional networks. Most similar to our architecture is FNet (Lee-Thorp et al., 2022), which replaces self-attention with a standard Fourier transform and achieves competitive performance with the transformer encoder. Rao et al. (2021) introduces GFNet, which similarly replaces self-attention with spectral operations but focuses on the vision domain. Patro et al. (2023) propose Spectformer, a vision transformer architecture that incorporates both spectral and attention layers. Other works use Fourier analysis to understand what functions neural networks learn, noting a preference for functions with low sensitivity (Rahaman et al. (2019), Xu et al. (2019), Xu (2020), Vasudeva et al. (2024)). Our work provides another justification for applying operations in the frequency domain, as MLPs trained on language modeling appear to develop a strong bias towards low-frequency computations.

**Training models to find implicit bias**    Closely related to our work is Bachmann et al. (2023), which investigates how lack of inductive bias can be compensated with increased scale and compute. However, Bachmann et al. (2023) focuses on scaling laws for MLPs on vision tasks, whereas we scale MLPs as a basis for studying what structures emerge as natural biases for language modeling. Also similar to our work is Neyshabur (2020), which explores how to use a learning algorithm with bias for sparse connectivity to replace the inductive bias of convolutions in vision tasks. Additionally, Fernando et al. (2016) proposes an evolution-based algorithm to learn convolutions. Towards understanding the impact of architectural bias, d'Ascoli et al. (2019) compare the bias of a CNN to a fully connected network through the lens of training dynamics and show the benefits of the CNN prior in achieving good generalization.

# 7  DISCUSSION

The deep learning community has increasingly aimed to train versatile models at the largest possible scale. We have pushed this effort to the extreme by training simple MLPs on large scale language modeling datasets. We demonstrate that MLPs perform nontrivially as language models when provided with enough data and compute. They learn to exploit structure in the data through local "$n$-gram" neurons and global "bag-of-words" neurons. We find that the Discrete Cosine Transform can be used to unify these neuron types, and leverage this finding to develop an architecture that is not dependent on sequence length in its number of parameters. We leave this work as a first step in understanding and leveraging emergent structures to develop scalable and efficient architectures for language modeling.

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

## A  EXAMPLE LANGUAGE MODEL GENERATIONS

Below we list example generations from three models for qualitative comparison. The last four prompts are meant to probe the model's ability to use the context beyond the immediate last word in predicting the next token. All three models are trained on 10 billion tokens. We limit the generation to 64 tokens for each response. Each generation is sampled with temperature 0.5.

### MLP 38M PARAMETERS (WIDTH 16384, CONTEXT LENGTH 15)

- **The capital of France is Paris. The capital of Italy is Rome. The capital of Germany is Berlin. The capital of Spain is** the capital of the world, with the city of the capital, the capital, and the people of the city. The city was built in the 15th century and was built in the 18th century. It is also the home of the famous Pondicherry Castle, which is home to the famous Alps.

- **It was the best of times, it was the worst of times, it was the age of wisdom, it was the age of foolishness, it was the epoch of** the great apocalypse, and the fact that it was a terrible occasion. It was a night of pure joy. It was a family event. It was a beautiful day for us. We were lucky to have been able to visit the most beautiful and beautiful villages in the world. The beautiful city of Khar

- **The baseball player stepped up to the plate, ready to hit a home run. The pitcher threw the ball, and the bat**ters hit the ground. Spencer has been a valuable tool in the game for a long time. The game is simple, and you can play it with your friends, family, or any other family. This is a great place to look for a new home that is ready to move in.

- **These nocturnal creatures are adept hunters, using echolocation to locate prey and navigate in the darkness. A bat**htub can also be used to make the house look more appealing. If you would like to see more of the bathroom designs, you can check out our gallery. The Bajaj Designs Gallery is a collection of photographs, texts, and photographs. The exhibition will provide an overview of the exhibition,

- **The robber pointed his gun at the cashier and demanded money. It was a bank** robbery, and the police had to pay the police. The couple was arrested in the case, and the police were called to the scene. The victim was struck by a car in the back of the head, a man who had been hiding in the room, was about to die. They were not a threat but

- **The river flowed gently through the forest, its waters sparkling in the sunlight. Along the river bank**, the lake, and the sea, the lake, and the lake. The lake is the birthplace of the lake, and the lake is the lake of the lake. The lake is the largest lake in the world, and is home to one of the world's largest and oldest golf clubs, and

### MLP 20M PARAMETERS (WIDTH 8192, CONTEXT LENGTH 15)

- **The capital of France is Paris. The capital of Italy is Rome. The capital of Germany is Berlin. The capital of Spain is** located in the south-east of the city of Rome. The city was designated as a city of the city of Pellica. It was the first of its kind in the world to be added to the list of the best Westerners. The "The Secret of the Dragon" was a family drama about the

- **It was the best of times, it was the worst of times, it was the age of wisdom, it was the age of foolishness, it was the epoch of** the world."

  "I can't believe that."

  "No, sir," said the old gentleman. "Yes."

"I am sorry, I don't know. I have a very bad feeling about the situation, but this is not the case. I'm not sure how I can

- **The baseball player stepped up to the plate, ready to hit a home run. The pitcher threw the ball, and the bat** struck the bat. When the game was finished, the team was able to make a second-half impact, and the team did not have a chance to make the playoffs. The only thing that is more interesting is that the season is going to be a lot of fun. The next step is to get

- **These nocturnal creatures are adept hunters, using echolocation to locate prey and navigate in the darkness. A bat**on is a snake, and the scorpion is a symbol of the world that has been told to do. The most important thing to remember is that the work is done in a way that is not a good thing. This is a good way to start your business. You can use your own customer service

- **The robber pointed his gun at the cashier and demanded money. It was a bank** robbery. The bank's general manager, who has been in the market for a long time. He also has a keen interest in the sport, including his career as a former manager and manager of the club, will be looking to add another player to the roster. The club is the only player in the

- **The river flowed gently through the forest, its waters sparkling in the sunlight. Along the river bank**, the island is a great place to take a dip in the water. It is a very popular tourist attraction. The city is famous for its unique architecture and its abundant history, and its interesting history, its history, and its history. The Italian Renaissance: A new history of the city, the city of

## GPT2 20M PARAMETERS (WIDTH 8192, CONTEXT LENGTH 15)

- **The capital of France is Paris. The capital of Italy is Rome. The capital of Germany is Berlin. The capital of Spain is** the capital of the Swiss city of Leipzig. The city of Geneva is home to several famous landmarks and attractions. The museum047 is a great place to visit. The city is the largest in the world, with a population of about 2,000,000. I am looking at the number of people who

- **It was the best of times, it was the worst of times, it was the age of wisdom, it was the age of foolishness, it was the epoch of** the great plague. It was the day of the year that the earth was filled with its water, and the sun was shining. The sun was shining, the sun was shining, the sun was shining, the sky was shining. autocrossing over the hill, a small red-headed man was killed

- **The baseball player stepped up to the plate, ready to hit a home run. The pitcher threw the ball, and the bat**ters walked in the second and third, and then they walked in the second. They had the ball and they were playing their very own game. The game was played on the same day, but the game was played on the same day, so I knew it was going to be a good match for the game

- **These nocturnal creatures are adept hunters, using echolocation to locate prey and navigate in the darkness. A bat** is a small creature with a small body, which is a small creature. The Annie's face was completely red. She turned around and looked around. "What's the matter?" she asked. "You're going to be fine. I'm going to have to get back to the drawing board.

- **The robber pointed his gun at the cashier and demanded money. It was a bank** robbery, and he refused to leave. In addition to the allegations, the court denied the allegations. The case was dismissed by the Supreme Court of the United States, which found that the state of Florida had a significant influence on the state of Florida's economy. The state of Florida has been known for its

- **The river flowed gently through the forest, its waters sparkling in the sunlight. Along the river bank**, you'll find a large pool of water and an abundance of sunscreen to keep your skin dry and free of any harmful UV rays. The sun protection is also important to protecting your eyes. It also helps in keeping your eyes healthy. More than that, you can use a lot of natural ingredients

## B   TRAINING DETAILS

We use a learning rate of 0.001 with a cosine learning rate schedule, 10% warmup, and minimum learning rate 1e-5. We also use gradient clipping and 0.1 weight decay. We use 1 or 4 Nvidia H100 GPUs to train our models, with runtime varying between 1 and 4 GPU days. We use a batch size of 16,384 samples.

## C   ADDITIONAL DETAILS ON DCT-PARAMETERIZATION

### C.1   NOTES ON THE DCT-NEURON

Observe that since the DCT operation is linear, the DCT Neuron is a linear function of the input, i.e. this is just a re-parameterization of the "standard" neuron. Additionally, each neuron has $k \cdot d$ learned parameters (in our experiments we choose $k$ to be a small constant, e.g. $k = 4$), so the number of parameters does not depend on the sequence length $T$. Finally, we note that the DCT Neuron can express both the $n$-gram and the bag-of-words neurons: if we choose $t = k$, then $\text{DCT}^{(k)}$ operation only changes the basis of the linear space, but the neuron can express any linear function over the last $k$ tokens; alternatively, if $t = T$, then the neuron can express a low-frequency function (with only $k$ frequencies) over the entire sequence (for $k = 1$ the neuron computes a linear function of the average of sequence length). A linear combinations of such neurons (which still gives a linear function of the input) gives "hybrid" neurons, and choosing other values of $t$ (e.g., $t = T/2$) allows different "trade-offs" between sequence length and complexity.

### C.2   DCT-PARAMETERIZED ARCHITECTURE DESCRIPTION

We propose a DCT-Parameterization architecture by replacing the causal attention heads of the transformer with a causal DCT-head with hyperparameters $k$ (number of frequencies) and $t$ (look-back window size). The input to the DCT-head is a sequence of vectors $\boldsymbol{x}_1, \ldots, \boldsymbol{x}_T \in \mathbb{R}^d$. As in the transformer attention head, for efficiency reasons we begin by projecting each vector to a lower dimensional space, from dimension $d$ to dimension $d_h$, using the matrix $\boldsymbol{W}_{\text{proj}} \in \mathbb{R}^{d \times d_h}$. Denote the projection of the $i$-th vector by $\boldsymbol{x}'_i = \boldsymbol{W}_{\text{proj}}^{\top} \boldsymbol{x}_i$. Then, we define for every frequency a linear map, denoted $\boldsymbol{W}_1, \ldots, \boldsymbol{W}_k \in \mathbb{R}^{d_h \times d_h}$. The output at position $i$ of the DCT-head is given by: $\boldsymbol{o}_i = \sum_{j=1}^{k} \boldsymbol{W}_j^{\top} \text{DCT}_j^{(t)}(\boldsymbol{x}'_{i-t}, \ldots, \boldsymbol{x}'_i)$, where $\text{DCT}_j^{(t)}$ denotes the $j$-th frequency in the DCT operator over a signal of length $t$. So, each head has two hyperparameters $k, t$ and learned parameter matrices $\boldsymbol{W}_{\text{proj}}, \boldsymbol{W}_1, \ldots, \boldsymbol{W}_k$. Note that the number of parameters does not grow with the sequence length. As in the transformer architecture, we let $d_h = d/h$, where $h$ is the number of heads, and concatenate the outputs of all heads back to dimension $d$, followed by a linear layer, normalization and non-linearity.

### C.3   EXPERIMENTAL SETTING

We train the DCT-Parameterized MLP (DCT-MLP) on the C4 dataset (**?**). As the number of parameters of our MLP no longer depends on the context length, we train with a full context window size of 2048. We train DCT-MLPs of depth $6, 12, 18$, with $2 - 6$ Billion Tokens, and report the results on the evaluation data. We use 16 heads, using $k = 4$ for each head, and varying $t$ from 4 to 2048 (the maximal seqeunce length). We compare the performance to a GPT2-based transformer, with depth varying from 2 to 6. The results are shown in Figure 9. For each experiment we train the model on a single Nvidia-H100 GPU. Each training run takes between 5 to 38 hours, depending on the model size and number of tokens.

# D CONTEXT LENGTH SCALING FOR MLP MODEL

Below we plot how perplexity and number of parameters scales with context length in the MLP model, to highlight the impracticality of the MLP model for long context lengths as discussed in Section 2

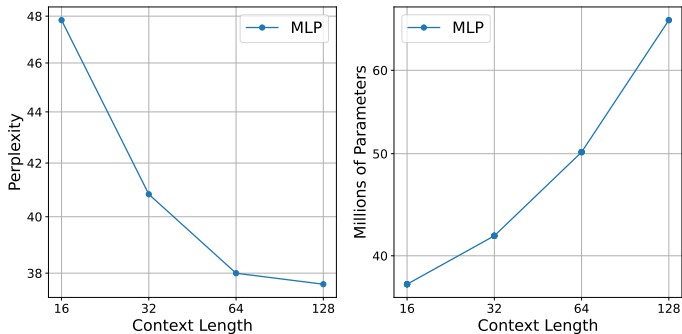

Figure 10: Context length scaling for a 16384-width MLP trained on 10b tokens. **Left:** Context length vs perplexity **Right:** Context length vs number of parameters

# E ADDITIONAL SETTINGS FOR NEURON TYPE CLASSIFICATION

We measure the proportion of neuron types over context length and model width. We find that the proportions are relatively consistent in these different settings, though large context length seems to have the greatest effect on the proportions.

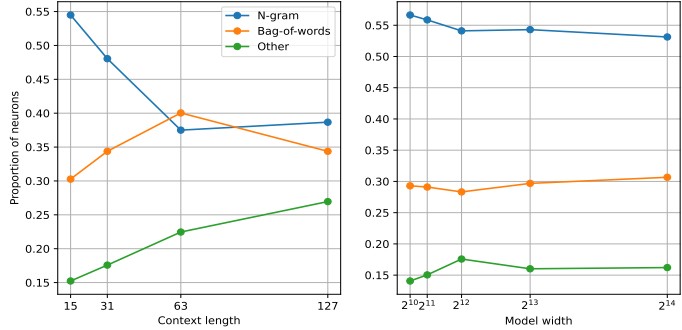

Figure 11: Proportion of neuron types over context length (Left) and model width (right), grouped into $n$-gram neurons, bag-of-words neurons, and other neurons for a 16384-width MLP trained on 10b tokens.

# F    ADDITIONAL EXAMPLES OF NEURONS

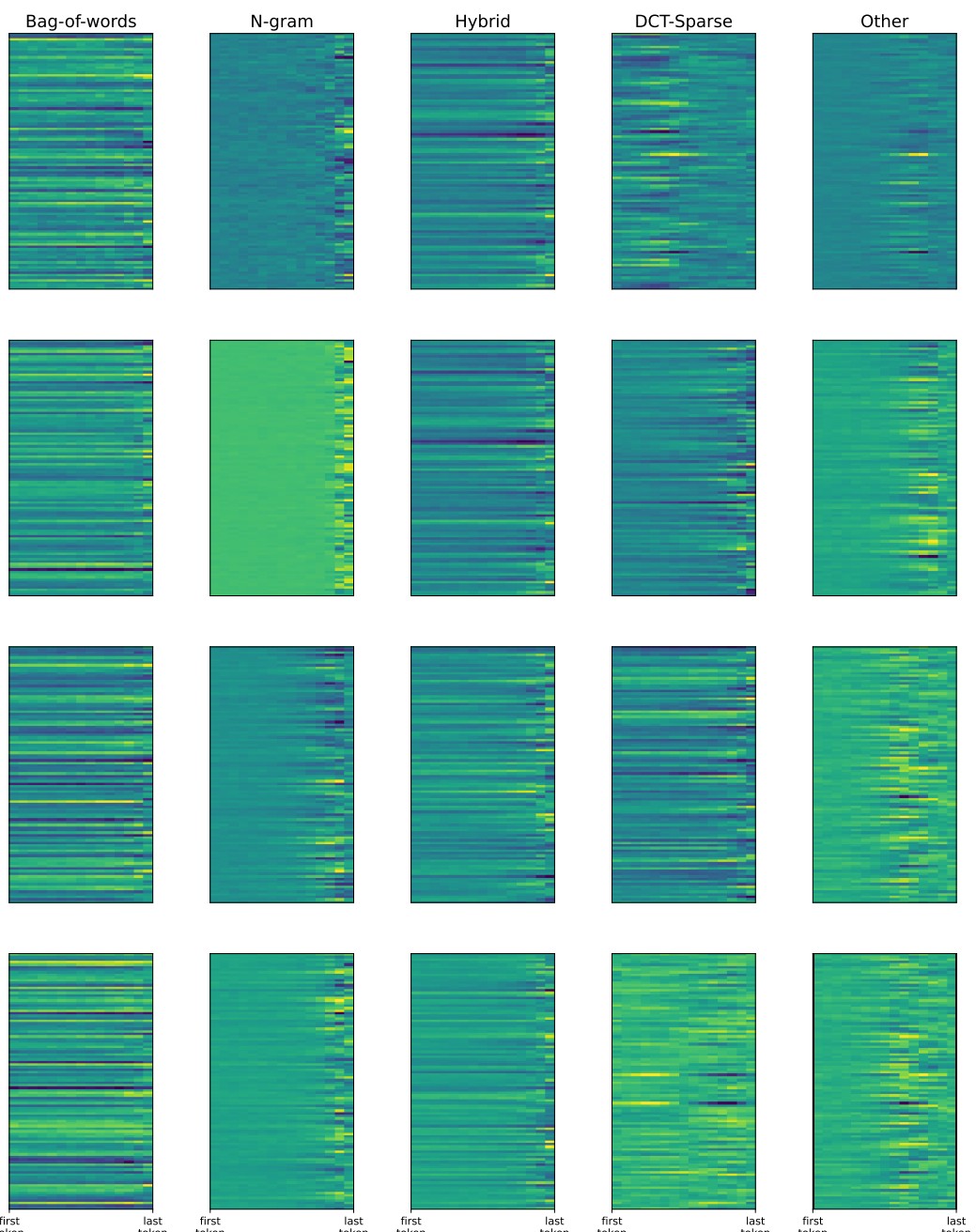

Figure 12: Examples of each type of neuron discussed. Neuron types are classified using the heuristics discussed in Section 4.1 and chosen at random. The neurons in the other category are chosen at random from the remaining neurons. For ease of visualization, we show the first 100 indices of the embedding dimension. All neurons are from a width 16384 MLP trained on 10b tokens.

# G  ADDITIONAL EXAMPLES OF KEYWORD ANALYSIS

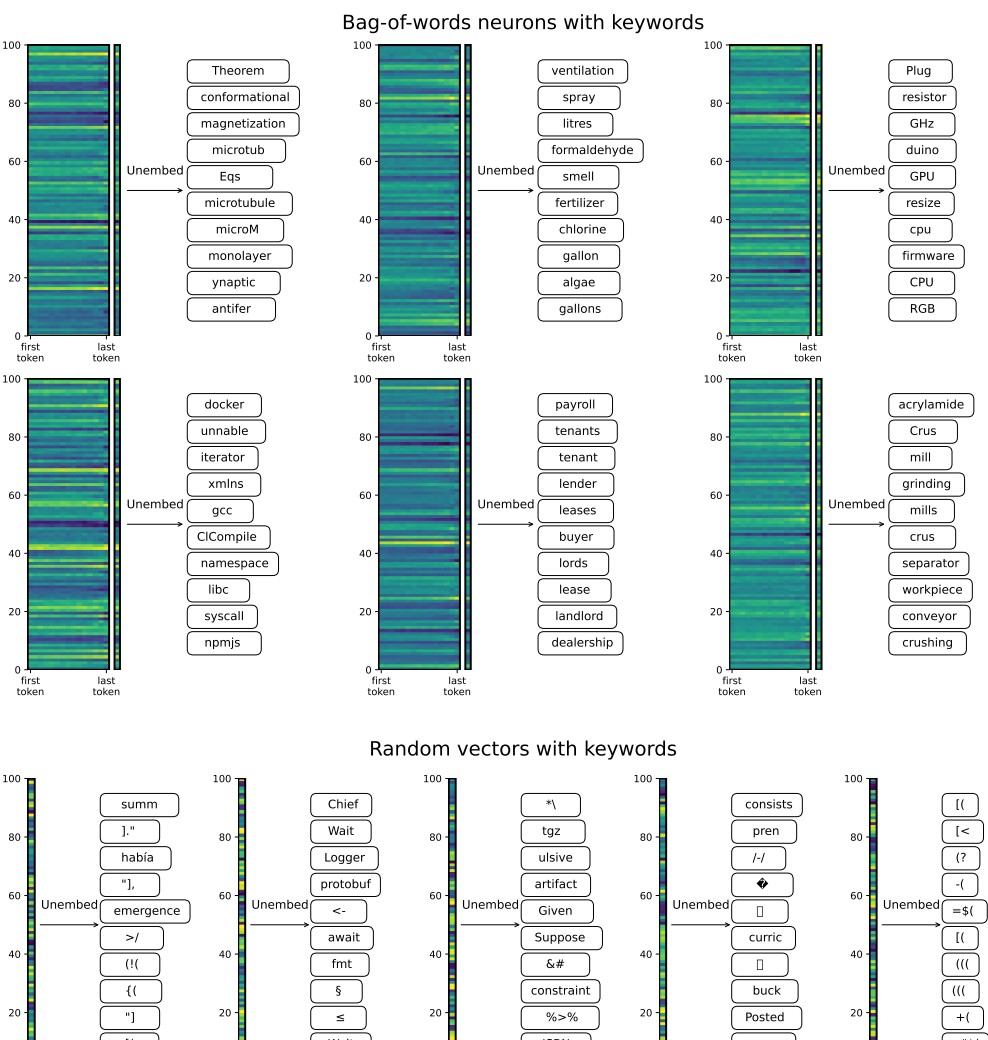

Figure 13: **Top:** Examples of associated keywords with bag-of-words neurons. The neurons illustrated are the six neurons with lowest variance according to the heuristic described in Section 4.1. For each neuron, we take the average across time and unembed the resulting vector to get the top-10 associated keywords. **Bottom:** As a baseline, we unembed randomly generated vectors $\in \mathbb{R}^d$ (normalized such that the entries sum to 1). For all neurons, only the first 100 rows of the embedding dimension are displayed.

N-gram neurons with keywords

Random vectors with keywords

Figure 14: **Top:** Examples of associated keywords with n-gram neurons. The neurons illustrated are the six neurons with the minimum norm across some index of time (as a modification of the heuristic described in Section 4.1 to derive a score). For each neuron, we take the column of time with the highest norm and unembed the resulting vector to get the top-5 associated keywords (first column of words). We project each keyword back into token space and run inference on our model with an input of all zeros besides the last token (which is the projected column). The second column of words for each neuron lists how the model completes each n-gram. **Bottom:** The same process is applied to four randomly generated vectors (normalized such that the entries sum to 1).

## H   DETAILS ON HEURISTICS USED TO CLASSIFY NEURONS

For a neuron $\boldsymbol{w}$ we compute

$$\text{ngram}(\boldsymbol{w}) := \left( \frac{1}{T} \sum_{i=1}^{T} \mathbf{1}_{\tilde{w}_i < t_{ngram}^{(l)}} \right) > p_{ngram} \text{ and } \exists i : \tilde{w}_i > t_{ngram}^{(h)}$$

$$\text{bag-of-words}(\boldsymbol{w}) := \frac{1}{d} \sum_{i=1}^{d} \text{var} \left( \frac{\boldsymbol{w}}{|\boldsymbol{w}|} \right) > t_{b\text{-}o\text{-}w}$$

where $\tilde{w}$ denotes $T$-dimensional vector obtained by taking the norm across the embedding dimension and the $\text{var}$ denotes the variance taken across the time dimension. We choose the parameters for the proportion of low norm time slices ($p_{ngram}$) and various thresholds ($t_{ngram}^{(l)}$, $t_{ngram}^{(h)}$, and $t_{b\text{-}o\text{-}w}$) based on visual inspection of the neurons and hold these values constant across models. For the included plots, we use the values $p_{ngram} = 0.6$, $t_{ngram}^{(l)} = 0.4$, $t_{ngram}^{(h)} = 0.9$, and $t_{b\text{-}o\text{-}w} = 0.0003$. For hybrid neurons, we relax the thresholds to $p_{ngram} = 0.5$, $t_{ngram}^{(l)} = 0.5$, $t_{ngram}^{(h)} = 0.8$, and $t_{b\text{-}o\text{-}w} = 0.0005$ and count neurons that are classified as both categories. For DCT-Sparse neurons, we use the same criteria as the $n$-gram neurons but on the neurons in the DCT-basis.

## I   $\ell_1/\ell_2$ MEASUREMENT IN FREQUENCY DOMAIN

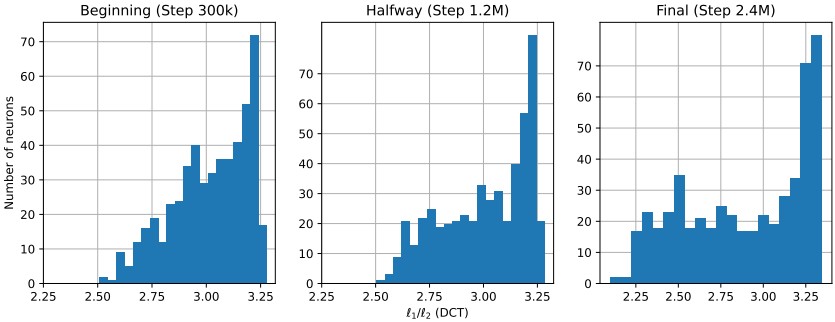

Figure 15: $\frac{\ell_1}{\ell_2}$ measured in the DCT representation of a 16384-width MLP trained on 10b tokens at initialization (right), halfway through training (center), and at the end of training (left). Inverse to the corresponding plot in the standard representation, the rightmost peak suggests n-gram neurons and the gradual accumulation of neurons in the left of the range suggests the emergence of bag-of-words neurons.

## J   EVOLUTION OF ROW NORMS OVER TRAINING

(a) Step 10,000

(b) Step 250,000

(c) Step 490,000 (Final)

Figure 16: Evolution of row norms of neurons over three checkpoints of a 16384-width MLP trained on 1b tokens.

