# OpenReview forum: "MLPs for NLP: Towards Discovering Inductive Bias From Scratch"
_ICLR.cc/2025/Conference — Submitted to ICLR 2025_

### Official Review · Reviewer_qZi5 · 2024-10-31

**Soundness:** 2
**Presentation:** 2
**Contribution:** 2
**Rating:** 3
**Confidence:** 4

**Summary:**

This paper studied applying the power of scale to the most straightforward neural network: multi-layer perceptron (MLP). Specifically, (1) the authors train MLPs to perform next-token prediction over 10 billion tokens. Results show that performance consistently improves with scale, though vanilla MLPs are inferior to those with Transformer architecture. (2) The authors also perform a mechanistic analysis of the trained models and identify a consistent emergent structure: bag-of-words and n-gram neurons.  (3) Using the discrete cosine transform, authors define a unified way of reparameterizing these neuron types such that the number of parameters per neuron does not depend on the sequence length.

**Strengths:**

This paper has some strengths.

(1) This paper applies a fresh perspective by examining the emergent neuron types in scaled-up MLPs and utilizing Discrete Cosine Transform (DCT) parameterization to address the fixed context limitation. Potentially, this research challenges the prevalent transformer paradigm, positioning MLPs as viable alternatives given sufficient scale and structural tuning.

(2) Identifying n-gram and bag-of-words neurons is a substantial contribution, providing a more interpretable foundation for language modeling tasks. The DCT-based parameterization adds a practical innovation by reducing parameter dependency on sequence length, which could influence future work on MLP-based sequence models.

**Weaknesses:**

Several weaknesses could probably be addressed.

(1) The structure of the paper could be improved to make the objectives and experimental logic clearer to readers. The authors first present experiments on scaling laws for MLPs, seemingly to provide a point of comparison with the scaling behaviors of transformer models. This comparison is a worthwhile angle, especially given the growing interest in understanding scaling laws across different architectures. However, the current writing could benefit from a more explicit framing of the experiment’s purpose—specifically, how it informs our understanding of inductive biases and scaling properties in different model types. Additionally, discussing how these findings fit into the broader research context on scaling laws might make the insights more impactful.

It is suggested that authors explicitly state the motivation, expected insights, and conclusions from the scaling law comparison between MLPs and transformers. It should also be clarified that how these results inform our understanding of inductive bias in model architectures.

(2) While the paper includes experiments on scaling laws and follows with analyses on neuron types and DCT, these sections currently feel somewhat disconnected. Clarifying the relationships between these components, or even providing a linking framework that explains the transitions and ties the findings together, could significantly strengthen the overall narrative. This would help in establishing a coherent story about how each part contributes to understanding inductive biases in MLPs and transformers alike.

It is suggested that authors add a paragraph explaining how the scaling law results motivate or inform the neuron type analysis and DCT parameterization.

(3) Revisiting the motivation behind this study, it is important to note that despite the demonstrated effectiveness of the transformer model, the authors have chosen to conduct a comprehensive analysis using a significantly simpler model architecture. In the Introduction, the authors assert that such experiments have the potential to enhance our understanding of the inductive biases that may be hardcoded and utilized in model design. However, the paper lacks substantial analysis or conclusions regarding this aspect. I appreciate the authors' trying towards this new and original perspective, however, the results currently obtained are far from sufficient to be a proper conference paper.

It is suggested that authors expand on how the observed emergent structures in MLPs (n-gram and bag-of-words neurons) could inform the design of inductive biases in more complex architectures like transformers.

(4) In addition to only observing the loss and perplexity, the work needs experiments from more perspectives to justify the effectiveness of its proposed DCT. The current lack of experiments makes it insufficiently convincing.

It is suggested to add comparisons on specific language tasks beyond next-token prediction, analyses of computational efficiency, or evaluations of the model's performance on different types of linguistic phenomena.

**Questions:**

(1) In this paper, the authors studied scaling laws, interpretability, and the identification of language models using a simpler architecture (MLPs). Do you think there are effective ways to connect these simple-architecture models with more complex transformer models, allowing us to transfer observations and analysis between them?

(2) In Figure 4, though authors successfully identify two important types of neurons---n-gram neurons and bad-of-word neurons---there still exist other types of neurons, which account for around 1/3 of all neurons. This could imply that analyzing the model using only the two types of neurons is not complete. How do you justify the completeness?

---

### Official Review · Reviewer_eVjp · 2024-11-02

**Soundness:** 3
**Presentation:** 3
**Contribution:** 3
**Rating:** 6
**Confidence:** 4

**Summary:**

The paper describes a study on using MLPs for language modeling instead of Transfomer networks. It analyses the resulting models and identifies several types of neurons which appear to function similar to count-based language models. The authors then use a discrete cosine transform-based neuron to train models which to use long context.

**Strengths:**

* Studying simpler network architectures is an interesting research direction
* Comparison to transformer networks and detailed analysis of the resulting models

**Weaknesses:**

* Section 4.3 mentions that the eauthors found neurons which resembled low frequency cosine waves. Can this be a little bit more discussed? Why was it cosine waves rather than something else? I am struggling to interpret the corresponding part in Figure 6.
* The paper identifies different types of neurons such as n-gram and bag-of-word type neurons but this does not appear to be unique to MLPs: self-attention layers are known to mean pool or spike on certain parts of the input. The current paper draws no relationship between this.
* The related work should cite some of the earlier comparisons between attention models and simpler approaches such as "Pay Less Attention with Lightweight and Dynamic Convolutions" by Wu etal.

**Questions:**

* Can you better motivate the use of DCT neurons?
* Can you draw comparisons between the neurons in MLP and attention layers?

---

### Official Review · Reviewer_YSfY · 2024-11-04

**Soundness:** 2
**Presentation:** 3
**Contribution:** 2
**Rating:** 3
**Confidence:** 4

**Summary:**

This paper explores the effectiveness of using Multi-Layer Perceptrons (MLPs) for NLP tasks, specifically focusing on next-token prediction in language modeling.

**Strengths:**

- By using a simple architecture, the authors reveal inherent patterns and inductive biases, particularly useful for understanding fundamental language structures that emerge without complex mechanisms like attention.
- The application of DCT offers a promising direction for creating efficient long-context architectures, possibly inspiring new hybrid architectures that combine the simplicity of MLPs with inductive biases from classical NLP methods.
- The paper provides detailed analyses on neuron types and the evolution of these types over training. This neuron-level analysis offers an intriguing perspective on how models develop interpretative structures from scratch.

**Weaknesses:**

- The experiments and comparisons are out of context compared to the latest research context of Transformer architectures.
- This paper largely repeats experiments in Bachmann et al. (2023). The novelty is questionable as there's less and less difference between vision and language.
- The MLP approach falls behind in computational and data efficiency compared to transformers.
- Although neuron types were categorized as n-gram and bag-of-words, the functional interpretability of these neurons remains a challenge.

**Questions:**

See "Weaknesses."

---

### Official Review · Reviewer_Urev · 2024-11-04

**Soundness:** 2
**Presentation:** 1
**Contribution:** 2
**Rating:** 3
**Confidence:** 3

**Summary:**

This paper train a large-sacle MLPs on MLP and attemp to explore the scaling laws of such models. The paper perform a mechanistic analysis of the model, showing most neurons in the first hidden layer either perform arbitrary linear functions over a small look-back window, or low-frequency functions over the entire context. The authors also have experiments using discreate cosine transform (DCT-Parameterized Neuron) to show the number of parameters per neuron does not depend on the sequence length.

**Strengths:**

- the authors design multiple experiments to analysis the indutive bias in modeling next word prediction

**Weaknesses:**

* The paper is not easy to follow. I believe I would have had a better understanding of the paper if the authors have time to polish the paper better. The paper shows eveveral interesting experiments, but the findings are not cohereent and not very well polished.

* the motivation is not clear to me. If the goal is to "Discovering Inductive Bias From Scratch", why choosing MLPs over other archtectures?

* maybe tighten the narrative.  The two major components of the paper, i.e. the mechanistic analysis and the cosine transforms are like a disjointed two sets to me.

**Questions:**

See weakness. Overall, the paper is very interesting, but there is a lot of room to improve the presentation.

---

### Meta-Review · Area_Chair_Hevq · 2024-12-22

**Metareview:**

The paper under review explores the use of Multi-Layer Perceptrons (MLPs) for language modeling tasks, particularly next-token prediction, and conducts analyses on the resulting models including investigations into neuron types and the application of Discrete Cosine Transform (DCT). It attempts to understand the scaling laws and inductive biases associated with MLPs as an alternative to the commonly used Transformer architectures. However, authors do not respond to the concerns raised by the reviewers. Based on the combined reviews, the paper currently does not meet the standards for acceptance.

**Additional Comments On Reviewer Discussion:**

No Discussion.

---

### Decision · Program_Chairs · 2025-01-22

Reject